# A Single-Cell Atlas of the Substantia Nigra Reveals Therapeutic Effects of Icaritin in a Rat Model of Parkinson’s Disease

**DOI:** 10.3390/antiox13101183

**Published:** 2024-09-30

**Authors:** Hao Wu, Zhen-Hua Zhang, Ping Zhou, Xin Sui, Xi Liu, Yi Sun, Xin Zhao, Xiao-Ping Pu

**Affiliations:** 1National Key Research Laboratory of Natural and Biomimetic Drugs, Peking University, Beijing 100191, China; 1310307406@bjmu.edu.cn (H.W.); 1611210107@bjmu.edu.cn (X.L.); sunyi@bjmu.edu.cn (Y.S.); 2Department of Molecular and Cellular Pharmacology, School of Pharmaceutical Sciences, Peking University, Beijing 100191, China; 3State Key Laboratory of Vascular Homeostasis and Remodeling, Peking University, Beijing 100191, China; 4School of Pharmaceutical Sciences, Sun Yat-sen University, Guangzhou 510006, China; zhangzhh68@mail2.sysu.edu.cn; 5Institute of Medicinal Plant Development, Chinese Academy of Medical Sciences & Peking Union Medical College, Beijing 100193, China; 2363190572@pku.edu.cn; 6The First Affiliated Hospital, Heilongjiang University of Chinese Medicine, Harbin 150040, China; suixin9558@gmail.com; 7China State Key Laboratory of Cell Biology, Center for Excellence in Molecular Cell Science, Chinese Academy of Sciences, Shanghai 200031, China

**Keywords:** Parkinson’s disease, rotenone, icaritin, single-cell RNA sequencing, neuroinflammation, oxidative stress

## Abstract

Degeneration and death of dopaminergic neurons in the substantia nigra of the midbrain are the main pathological changes in Parkinson’s disease (PD); however, the mechanism underlying the selective vulnerability of specific neuronal populations in PD remains unclear. Here, we used single-cell RNA sequencing to identify seven cell clusters, including oligodendrocytes, neurons, astrocytes, oligodendrocyte progenitor cells, microglia, synapse-rich cells (SRCs), and endothelial cells, in the substantia nigra of a rotenone-induced rat model of PD based on marker genes and functional definitions. We found that SRCs were a previously unidentified cell subtype, and the tight interactions between SRCs and other cell populations can be improved by icaritin, which is a flavonoid extracted from *Epimedium sagittatum* Maxim. and exerts anti-neuroinflammatory, antioxidant, and immune-improving effects in PD. We also demonstrated that icaritin bound with transcription factors of SRCs, and icaritin application modulated synaptic characterization of SRCs, neuroinflammation, oxidative stress, and survival of dopaminergic neurons, and improved abnormal energy metabolism, amino acid metabolism, and phospholipase D metabolism of astrocytes in the substantia nigra of rats with PD. Moreover, icaritin supplementation also promotes the recovery of the physiological homeostasis of the other cell clusters to delay the pathogenesis of PD. These data uncovered previously unknown cellular diversity in a rat model of Parkinson’s disease and provide insights into the promising therapeutic potential of icaritin in PD.

## 1. Introduction

The exact cause of neurodegenerative diseases, which are defined as common degenerative responses to diverse neurological processes, has not been fully elucidated [1]. Parkinson’s disease (PD) is a common neurodegenerative disease characterized by the preferential loss of dopaminergic neurons in the substantia nigra pars compacta (SNpc) and the presence of intracellular inclusions because of neuroinflammation and oxidative stress; moreover, its clinical symptoms include tremor, stiffness, and bradykinesia [2]. Although the loss of dopaminergic neurons is central to the pathology of Parkinson’s disease, the roles of other cell types, such as astrocytes and microglia, have also been proposed as contributing factors to fine motor control deficits via inducing neuroinflammation and oxidative stress [3,4]. The rat SNpc is sorely understudied compared with that of the mouse, for which single-cell/nuclei studies [5,6,7] have mapped and characterized the diverse cell-type populations and identified cell-type-specific disease associations. Thus, a systematic study of cell-type-specific gene expression in the SNpc of rats will further help us understand the selective vulnerability of dopaminergic neurons in PD and provide important insights into the potential contributions of SNpc cell types to other neurological disorders.

Identifying cell types linked to specific diseases is essential for understanding the causal mechanisms of molecular pathogenesis. PD is strongly influenced by genetic variations altering gene function [8]. By associating genetic variations with particular genes, we can map their spatial expression patterns and connect specific cell types to the disease. Although the cell types associated with genetic risk for a disease may not always be the ones most directly related to the defining symptoms, it is crucial to reassess our etiological hypotheses when obtaining data.

The classic experimental models of PD are established using toxic compounds, such as the pharmacological agent rotenone, which induces the selective degeneration of nigrostriatal neurons to reproduce the pathological and behavioral changes of the human disease in rodents [2]. Although the underlying mechanisms are not yet fully understood, oxidative stress, neuroinflammation, and abnormalities in energy metabolism are closely associated with the development of PD. A series of natural antioxidant compounds, such as kaempferol, icaritin, and artemisinin, have been confirmed by preclinical studies to effectively treat PD [9]. Due to their natural origins, they are expected to have a more favorable safety profile. Most of these compounds can restore energy metabolism and counteract oxidative damage, thereby exerting anti-PD effects. Our laboratory’s preliminary research has confirmed the beneficial therapeutic effects of icaritin in PD mice. Icaritin is produced via the hydrolysis of icariin, a flavonoid extracted from *Epimedium sagittatum* Maxim. and exerts anti-neuroinflammatory, antioxidant, and immune-improving effects in PD [10,11].

Here, after icaritin treatment, we used single-cell RNA sequencing (scRNA-seq) combined with in situ imaging techniques to characterize the role of specific neurons in a preclinical rotenone-induced rat model of PD. Our analyses led to the identification of eight neuronal subsets, among which cluster 5 comprised mostly dopaminergic neurons. Furthermore, a previously unknown cell subtype was identified in the substantia nigra as a synapse-like cell cluster, i.e., synapse-rich cells (SRCs). In addition, SRCs were significantly reduced in the substantia nigra of rats with PD, which could be reversed by icaritin treatment. We also observed that icaritin application improved neuroinflammation, oxidative stress, and survival of dopaminergic neurons and promoted the recovery of abnormal energy metabolism, amino acid metabolism, and phospholipase D metabolism in astrocytes from rats with PD using matrix-assisted laser desorption/ionization–mass spectrometry imaging (MALDI–MSI). These data provide insights into the promising therapeutic potential of icaritin in PD.

## 2. Materials and Methods

### 2.1. Reagents and Drugs

Rotenone was purchased from Sigma-Aldrich (St. Louis, MO, USA). Selegiline Hydrochloride Tablets (approval number: H20160342) were produced by Orion Corporation (Espoo, Finland). Nano icaritin aqueous solution (production batch no.: 11001) was produced by Lunan Pharmaceutical Group (Linyi, China) (Chemical structure of icaritin, Figure 1A). Mouse antibodies against glycosylphosphatidylinositol-specific phospholipase D (GPI-PLD, sc-365096) were obtained from Santa Cruz Biotechnology, Inc. (Dallas, TX, USA).

### 2.2. Animals Care

Eighty male age-matched Sprague Dawley rats weighing 230–260 g were purchased from Beijing Vital River Laboratory Animal Technology Co., Ltd. [Beijing, China; license no.: SCXK (Beijing) 2016-0011]. The rats were labeled, weighed, and caged; water and food were provided ad libitum. The animals were maintained at 22–24 °C under a 12 h light/dark cycle and 50–60% humidity for 1 week for acclimatization before treatment. All animal experiments were approved by the Peking University Biomedical Ethics Committee (Beijing, China; approval no. LA2017282) and were conducted by experimenters who had an employment certificate from the Department of Laboratory Animal Science, Peking University Health Science Center, China. All efforts were made to minimize animal suffering.

### 2.3. Rotenone Rat Model of PD

Rats were randomly divided into six groups: control (Control), PD model (model), selegiline (selegiline, 10 mg/kg), low-dose icaritin (icaritin_L_, 3.27 mg/kg), medium-dose icaritin (icaritin_M_, 6.54 mg/kg), and high-dose icaritin (icaritin_H_, 13.08 mg/kg) [10]. Nano icaritin aqueous solution (production batch no.: 11001) was produced by Lunan Pharmaceutical Group (Shandong, China) (Chemical structure of icaritin, Figure 1A). One week before rotenone treatment, the rats were intragastrically administered the indicated doses of selegiline or icaritin or an equal volume of purified water for 18 days. To establish the PD model, 2.5 mg/kg rotenone, which was prepared by dissolving 250 mg of rotenone in 2 mL of dimethyl sulfoxide and 98 mL of olive oil, was administered subcutaneously for 11 consecutive days [12,13]. The drug was intragastrically administered 1 h before rotenone injection, and the control and PD model groups were subcutaneously administered an equal volume of dimethyl sulfoxide/olive oil mixture (2:98). One day after the last injection, behavioral assessments were performed using the rearing behavior and wire grip tests. On the 27th day, all rats were sacrificed for further evaluation. The total number of animals in the control, model, selegiline, icaritin_L_, icaritin_M_, and icaritin_H_ groups was 10, 14, 14, 14, 14, and 14, whereas the number of surviving animals before sacrifice was 10, 13, 12, 12, 13, and 12, respectively. The body weight of all rats was recorded daily after treatments (Appendix A). All rats were humanely killed (via pentobarbital overdose), and samples were collected for further analysis.

### 2.4. Behavioral Assessments

#### 2.4.1. Rearing Behavior Test

Each rat was placed in a clear Plexiglass cylinder (height, 30 cm; diameter, 20 cm) for 5 min to quantify the number of rears [14]. The rats were classified as rear when they raised the forelimbs higher than the horizontal line of the shoulders and contacted the inner barrel wall using the unilateral or bilateral forelimbs. Removal of the bilateral forelimbs from the wall and contact with the bottom was required before scoring another rear.

#### 2.4.2. Wire Grip Test

The forelimb grip was evaluated in rats that were suspended by their forepaws on a horizontal metal rod with a diameter of 4 mm at approximately 12 cm above a padded floor [13]. Each rat was timed for 5 min, and the latency to fall was recorded.

### 2.5. Measurement of Dopamine, Serotonin, and Their Metabolites in Striatum

The levels of dopamine and its metabolites (3,4-dihydroxyphenylacetic acid and homovanillic acid), as well as those of serotonin and its metabolite (5-hydroxyindoleacetic acid) were assessed in the striatum of rats by high-performance liquid chromatography using an electrochemical detector (BAS LC-4B, BASi, West Lafayette, IN, USA), as described previously [15]. The composition of the mobile phase was 0.1 M acetate–citrate buffer (pH 3.7) containing 15% methanol, 1.09 mM octyl sodium sulfate acid, 0.4 mM dibutylamine, and 0.2 mM EDTA. The flow rate in the reversed-phase column was 1.2 mL/min at 25 °C (Dikma, diamond, C-18 ODS, 2504 mm, Beijing, China). After centrifugation (4 °C, 20,000× *g*, 20 min), 70 μL of the supernatant of the striatal tissue homogenate was injected directly into the high-performance liquid chromatography system. Data were calibrated using an external standard. The levels of DA and serotonin, as well as those of their metabolites, were calculated and expressed as μg/g of wet tissue weight.

### 2.6. Western Blotting Analysis

Midbrain tissue samples of the rats were lysed in RIPA lysis buffer as described previously [16], and protein aliquots (30 μg) were separated using 10% sodium dodecyl sulfate–polyacrylamide gel electrophoresis (SDS–PAGE). After separation, the proteins were electrophoretically transferred onto polyvinylidene fluoride membranes (catalog # IPVH00010; Millipore, Bedfordshire, PA, USA). The membranes were then blocked with 5% skim milk in Tris-buffered saline containing 0.1% Tween 20 (catalog # T104863; Aladdin, Shanghai, China) and incubated overnight at 4 °C with the rabbit antibody against GPI-PLD (1:1000). Finally, the membranes were incubated with a goat peroxidase-conjugated secondary antibody and visualized using enhanced chemiluminescence (Elpis-Biotech, Lexington, KY, USA). Band intensities were quantified using ImageJ 1.52a (National Institutes of Health, Bethesda, MD, USA).

### 2.7. Isolation of the Substantia Nigra in Rats with PD and Cell Dissociation

All rats were injected with a small amount of ink at the anterior fontanel, and the brain was collected by craniotomy. The brain was then cut between −4.80 mm and −5.80 mm from the anterior fontanel, according to the fifth edition of the *Stereotactic Atlas of the Rat Brain*. The dark substantia nigra was identified, and the tissue was collected into a centrifuge tube, which was quickly frozen in liquid nitrogen. A single-cell suspension of each area was obtained by enzymatic digestion using a protocol described previously [17]. The overall cell viability was >90%, confirmed using the trypan blue exclusion test. The single-cell suspensions were counted using a Countess II Automated Cell Counter, and the cell density was adjusted to 700–1200 cells/μL before single-cell analysis.

### 2.8. Single-Cell RNA-Seq Library Preparation and Sequencing

The Seurat R package was applied to convert scRNA-seq data into Seurat objects in our study. Cell-level quality control was performed to filter cells based on (1) a total UMI count below 1000; (2) a gene number below 200; or (3) a mitochondrial gene percentage greater than 20%. The expression level of each gene in each cell was normalized with the NormalizeData function using the LogNormalize method and a scale factor of 10,000 to remove the effect of sequencing library size, which converted the expression values from UMI counts to ln [10,000 × UMI counts/total UMI counts in cell + 1]. All individual samples were integrated into Seurat using the canonical correlation analysis (CCA) pipeline to remove batch effects. The “SelectIntegrationFeatures” function was applied to select the features ranked according to the number of datasets in which they were detected. Next, the “FindIntegrationAnchors” function selected 2000 anchors between different samples using the top 50 dimensions from the CCA to specify the neighbor search space. “IntegrateData” was then applied to integrate the datasets using the pre-computed anchors, and the integrated dataset was scaled using “ScaleData”. A PCA and UMAP dimension reduction based on the top 20 principal components was performed. The identified clusters were visualized on the 2D map produced using the *t*-distributed UMAP method. The cells were then clustered using Seurat’s FindNeighbors with dimensions of 1–20 and FindClusters with a resolution of 0.5. Moreover, we used the FindAllMarkers functions for the detection of gene expression markers. Subsequently, we employed the SingleR package and the CellMarker dataset to annotate the cell types identified in our study.

The SubsetData function was also used to extract subclusters for downstream analysis. After the detection of clusters and gene expression markers in subclusters using the FindClusters and FindAllMarkers functions, a UMAP analysis was performed using the RUNUMAP function. The subclusters were annotated as described above.

### 2.9. GSVA and SCENIC Analysis

To perform the Gene Set Variation Analysis, the GSEABase package (version 1.44.0) was used to load the gene set file, which was downloaded and processed from the Kyoto Encyclopedia of Genes and Genomes database (https://www.kegg.jp/) (accessed on 25 May 2022). To assign pathway activity estimates to individual cells, we applied GSVA using the standard settings implemented in the GSVA package (version 1.30.0). The differences in pathway activities (scored per cell) were calculated using the LIMMA package (version 3.38.3).

The SCENIC analysis was run using the motifs database for RcisTarget and GRNboost (SCENIC version 1.1.2.2, which corresponds to RcisTarget 1.2.1 and AUCell 1.4.1) with default parameters. More specifically, we identified transcription factor (TF)-binding motifs that were over-represented on a gene list using the RcisTarget package. The activity of each group of regulons in each cell was scored using the AUCell package. To evaluate the cell type specificity of each predicted regulon, we calculated the regulon specificity score based on the Jensen–Shannon divergence, which is a measure of the similarity between two probability distributions. Specifically, we calculated the Jensen–Shannon divergence overlap between each vector of binary regulon activity and the assignment of cells to a specific cell type. The connection specificity index for all regulons was calculated using the scFunctions (https://github.com/FloWuenne/scFunctions/) (accessed on 25 May 2022) package.

### 2.10. Analysis of Cell–Cell Interactions

CellChat in the R package was used to perform a cell–cell communication analysis (http://www.cellchat.org/) (accessed on 25 May 2022). Briefly, based on manually curated databases that consider known structural compositions of ligand–receptor interactions, CellChat infers and analyzes intercellular communication networks from scRNA-seq data using network analysis and pattern recognition. A Seurat object, including the count matrix and clustering results from each dataset, is imported into CellChat. The default human database was used for single-cell dataset analyses.

### 2.11. Molecular Docking

Molecular docking analyses were performed using MOE v2019.1 [18]. The 3D structure of icaritin was downloaded from the PubChem database, whereas the 3D structures of the five target proteins were downloaded from the UniProtKB database (BCL11A [Uniprot ID: D3ZSY3], CEBPB [Uniprot ID: P21272], NR3C2 [Uniprot ID: P22199], SMAD3 [Uniprot ID: P84025], and TCF4 [Uniprot ID: Q62655]). Prior to docking, the AMBER10:EHT forcefield and implicit solvation model of the Reaction Field were selected.

The MOE-Dock program was used for molecular docking simulations between small molecules and their targets. The docking workflow followed the “induced fit” protocol in which the side chains of the receptor pocket are allowed to move according to ligand conformations, with a constraint on their positions. The weight used for tethering side chain atoms to their original positions was 10. For each ligand, all docked poses were first ranked based on the London dG scores, and a forcefield refinement was performed for the top 20 poses, followed by rescoring using GBVI/WSA dG. Molecular graphics were generated via MOE.

### 2.12. Matrix-Assisted Laser Desorption/Ionisation–Mass Spectrometry Imaging (MALDI-MSI)

After the completion of behavioral assessments, three rats each from the control, model, and icaritin groups were euthanized via intraperitoneal injection of a 3-fold anesthetic dose of 0.5% pentobarbital sodium, and their brains were rapidly excised, snap frozen in liquid nitrogen, and stored at −80 °C.

Complete and smooth transverse frozen brain slices (thickness, 10 μm) were prepared using a cryostat microtome (Scotsman Jencons, Nussloch, Germany) at −17 °C. The slice position was −5.28 mm away from the bregma in rats [19]. Subsequently, the tissue slices were transferred onto indium tin oxide-coated glass slides (Bruker Daltonics, Bremen, Germany) and desiccated using a vacuum pump for 30 min before matrix spraying. Then, the tissue slices were sprayed using an ImagePrep tissue imaging matrix sprayer (Bruker Daltonics, Billerica, MA, USA).

The slices were analyzed via MALDI–MSI using an Autoflex Speed™ MALDI TOF (TOF) system equipped with a 2 kHz Smartbeam-II laser (Bruker Daltonics, Bremen, Germany), according to the protocols reported in previous studies [20,21]. The results of MALDI–MSI were analyzed according to the study of Liu et al. [21].

### 2.13. Statistical Analysis

Data are expressed as mean ± SEM. Statistical analyses of MALDI–MSI data were performed using SCiLS Lab based on the normalization of total ion chromatography data. All data were analyzed using GraphPad Prism 8.0 (GraphPad Prism, San Diego, CA, USA). All results were compared using one-way analysis of variance. Comparisons between groups were performed using Fisher’s LSD test. A *p*-value of <0.05 was considered to indicate statistical significance.

## 3. Results

### 3.1. Establishment of the Rat Model of PD and Single-Cell Transcriptome Atlas of Cells in the Substantia Nigra

Patients with PD exhibit symptoms such as tremors, stiffness, and bradykinesia. Therefore, we evaluated whether rats exhibited behavioral signs of PD after rotenone injection and assessed the efficacy of icaritin. The daily intragastric administration of icaritin (Figure 1A) was initiated 7 days before the rotenone treatment and was continued for 11 days after its initiation (Appendix A). The rats were sacrificed on day 2 after the appearance of dopaminergic lesions. We demonstrated that rotenone-treated rats showed deficits in rearing behavior and wire grip tests (Figure 1B), all of which were reversed by treatment with icaritin or selegiline. Furthermore, a decrease in body weight was observed after the subcutaneous injection of rotenone owing to its toxic effects (Appendix A). Collectively, these results indicated that icaritin significantly improved the rotenone-induced behavioral abnormalities in rats. Behavioral disorders are closely related to the degradation of dopaminergic neurons because of neuroinflammation and oxidative stress [7]; therefore, we assessed the levels of dopamine (DA), serotonin, and their metabolites in the striatum. Treatment with icaritin or selegiline partially reversed the reductions in the levels of DA, 3,4-dihydroxyphenylacetic acid (DOPAC, a DA metabolite), and serotonin in the striatum of rotenone-treated rats, demonstrating that icaritin improved the exercise capacity in the rat model of PD by inhibiting DA, DOPAC, and serotonin and that the medium dose of icaritin used in this study was most effective (Figure 1C). Therefore, a medium icaritin dose of 6.54 mg/kg was selected for analyzing the effect of icaritin, and the 5-HIAA level has not changed (Appendix A).

To further examine the role of icaritin in the substantia nigra, we applied scRNA-seq to derive a detailed transcriptome of the cells in the substantia nigra of rats with PD after icaritin treatment. We collected samples of the substantia nigra of rats according to the fifth edition of the “Rat Brain Stereotaxic Atlas” (Figure 1D). After digestion with enzymes, cell suspensions with stringent quality control (Appendix A) were sequenced using a 10× genomics system. We obtained 14,123 cells in total, which were consistently distributed in the UMAP plot. All substantia nigra cells were sequenced at a high sequencing depth (60,000 reads per cell), with nearly 1500 genes detected per cell. We obtained 20 clusters of cells from the whole substantia nigra (Appendix A). A differential gene enrichment analysis revealed that clusters 0, 1, and 9 and clusters 2, 4, 10, and 15 were similar (Appendix A). Oligodendrocytes and excitatory neurons were the main components of the substantia nigra (Appendix A). Therefore, all substantia nigra cells were further classified into the seven types shown in the dot plot and the heatmap (Figure 1E–G and Appendix A). The top markers could be identified based on genome alignment using CellRanger, involving classic marker genes [22], such as *Mog*, *Hapln2*, and glutamate metabotropic receptor 3 (*Grm3*) gene for oligodendrocytes (ODCs); *Snhg11*, *Ntng1*, and *Lrrc7* for neurons; *Slc4a4*, *Agt*, and *Gja1* for astrocytes; *Pdgfra*, *Arhgap31*, and *Vcan* for oligodendrocyte progenitor cells (OPCs); *Ctss*, *Arhgap15*, and *C1qa* for microglia; *Ryr2*, *Cdh9*, and *Gria1* for synapse rich cells (SRCs); and *Flt1*, *Slc6a20*, and *Cldn5* for endothelial cells (ECs). After rotenone or icaritin treatment, the substantia nigra of rats retained all seven cell clusters, with a special cluster, SRCs, being significantly reduced in the model group compared with the control and icaritin groups (Figure 1H,I).

### 3.2. Cells Characterized by Specific Synapse-Related Genes Are Significantly Reduced in the Substantia Nigra of Rats with PD and Recovered by Icaritin Treatment

Previous research revealed that cell populations in the substantia nigra mainly included neurons, oligodendrocytes, vascular leptomeningeal cells, and astrocytes without any SRCs [23]. In our study, the number of SRCs in the model group was significantly lower than that in the control and icaritin groups (Figure 2A). The marker genes of SRCs included ryanodine receptor 2 (*Ryr2*), calcium-dependent cell–cell adhesion molecules (*Cdh9*), glutamate ionotropic receptor AMPA type subunit 1 (*Gria1*), EPH receptor A7 (*Epha7*), and Anoctamin 3 (*Ano3*), etc. (Figure 2B). Among them, *Ryr2* and *Cdh9* were highly specifically enriched in SRCs (Figure 2C). Besides, RYR2 and CDH9 were confirmed to be expressed in the rat substantia nigra and distinguished from tyrosine hydroxylase via immunostaining, suggesting that SRCs may be a new cell population present in the substantia nigra of rats (Figure 2D,E). Moreover, differential gene expression (DGE) analysis showed that genes associated with synapse organization and cellular, molecular transport-related functions were significantly expressed in SRCs, indicating that this cell population is mainly involved in the intracellular transport of synaptic neurotransmitters, metal ions, and vesicles (Figure 2F,G). Furthermore, the Kyoto Encyclopedia of Genes and Genomes analysis revealed that dopaminergic synapses, glutamatergic synapses, GABAergic synapses, long-term potentiation-related pathways, and the Phospholipase D signaling pathways were significantly enriched in SRCs (Figure 2H). Thus, the SRC population was specifically characterized by synapse-related genes in the rat substantia nigra and was significantly reduced in PD, which could be reversed by treatment with icaritin.

### 3.3. SRCs Interact Tightly with Other Cell Populations in the Substantia Nigra of Rats with PD

To further characterize the SRCs, the transcription factors (TFs) featured among all clusters were determined, such as SMAD family member 3 (SMAD3), the BAF chromatin remodeling complex subunit (BCL11A), CCAAT/enhancer-binding protein beta (CEBPB), nuclear receptor subfamily 3 group C member 2 (NR3C2), and transcription factor 4 (TCF4) were identified exclusively in SRCs (Figure 3A–C).

Furthermore, these TFs interacted with each other to jointly regulate the expression of a large number of genes (Appendix A), including genes related to synaptic plasticity, such as *Ptn* and *Afap1l2*. Moreover, a molecular docking experiment further demonstrated that icaritin bound to these TFs (Appendix A and Table 1), indicating that SRCs have enhanced synaptic characterization. In addition, ligand–receptor interaction analyses of the seven cell clusters demonstrated changed cell–cell interactions among the cell types (Figure 3D,E). In particular, the SRCs had apparent interactions with neuron clusters. Genes related to the formation or maintenance of synaptic junctions, i.e., *Nrxn1* and *Nrxn3*, interacted most significantly with the synaptic-signal-transmission-related gene *Nlgn1*. *Nrxn1* mutations cause a variety of neurological diseases, including attention deficit hyperactivity disorder, intellectual disability, seizures, schizophrenia, and mood disorders [24], suggesting its important role in the nervous system. Other cell populations also interacted with the SRC population, such as oligodendrocytes and the synaptic plasticity; and learning-related behavior-related gene, *Ptn,* controlled oligodendrocyte precursor cell differentiation by enhancing the phosphorylation of *Afap1l2* to activate the PI3K–AKT pathway [25,26] (Figure 3F–H). However, the mechanism via which SRCs play a role in PD warrants further study.

### 3.4. Icaritin Treatment Improves Neuroinflammation, Oxidative Stress, and Synaptic Vesicle-Mediated Transport-Related Pathways in Neurons of the Substantia Nigra from Rats with PD

Neurons are one of the most important cell clusters in the substantia nigra [5,6,7]. The 3282 neurons were detected and subclustered to eight distinct subtypes based on relatively highly expressed genes, such as *Grid2* and *Pld5* for the neuron-0 subcluster, *Fstl4* and *Shisa6* for neuron-1, *Necab1* and *Ptpn3* for neuron-2, *Plp1* and *Apod* for neuron-3, *Foxp2* and *Angpt1* for neuron-4, *Ak5* and *Ptprd* for neuron-5, *Oxr1* and *Dgkb* for neuron-6, and *Apoe* and *Atp1a2* for neuron-7 (Figure 4A,B, Appendix A). To further characterize their functions, we compared gene pathway activities with specific phenotypic features and found that dopaminergic neurons were more likely to be the neuron-5 subcluster (Figure 4C). Moreover, pathways related to protein localization and synaptic vesicle-mediated transport were highly activated in the eight neuronal cell subclusters (Figure 4D). As shown in Figure 4E and Appendix A, enrichment analysis revealed that neuron-5 displayed high expressions of other genes involved in oxidative phosphorylation (such as *Ak5*), IL-6/JAK/STAT3 pathway, and PI3K/AKT signaling, suggesting that inflammation and oxidative stress were strongly activated in these cells. Although neuron-2 and neuron-4 both exhibit lower oxidative phosphorylation and higher Wnt/β-catenin signaling, neuron-4 shows strong angiogenesis signaling, while neuron-2 displays a greater inflammatory response. Moreover, neuron-7 exhibited upregulated angiogenesis, adipogenesis, hedgehog signal, and interferon α/γ response. Furthermore, DGE analysis of the model and icaritin groups showed that the expression of several genes and TFs were significantly altered in all subclusters (Figure 4F,G). The expression of *Ttr*, which encodes transthyretin (a neuronal stress biomarker), was upregulated in all subclusters. Consistently, according to a previous study, the expression of *TTR* is upregulated in patients with neurodegenerative disorders [27]. The neuron-0 and neuron-7 subclusters with high expressions of the promelanin concentrating hormone (*Pmch*) gene might be related to motor behavior, considering PMCH as a neuromodulator in a broad array of neuronal pathways directed toward the regulation of goal-directed behavior [28]. In addition, neuron-5 with upregulated *Grm8*, *Adgrl2*, and *Tcf7* possibly played a role in neuroinflammation because *Grm8* protects against neurodegeneration during CNS inflammation and oxidative stress [29], *Adgrl2* is involved in the regulation of exocytosis, and *Tcf7* is related to inflammation development and neural defenses against virus infection. In contrast, the expression of *Grm8* was downregulated in neuron-4, suggesting a functional difference between the neuron-4 and dopaminergic neuron-5 subclusters in PD. The upregulation of ATPase H^+^-transporting accessory protein 2 (*Atp6ap2*) expression indicated that the ERK1/2 and Wnt pathways were activated in neuron-4 [30,31]. Taken together, these results indicate that icaritin significantly alters the expression of genes involved in neural development, survival, and maturation of dopaminergic neurons, neuroinflammation, and neuronal exocytosis in PD.

### 3.5. Amelioration of Icaritin on the Energy Metabolism, Phospholipid Metabolism, and Amino Acid Metabolism-Related Pathways in Astrocyte of the Substantia Nigra from Rats with PD

The 2462 astrocytes were clustered in five separate subsets (Figure 5A). The top markers could be identified using CellRanger for classic marker genes [22], such as *Plcb1* and *Trpm3* for the astrocyte-0 subcluster, *Snhg11* and *syt1* for astrocyte-1, *Pex5l* and *Tmeff2* for astrocyte-2, *Vim*, *Nckap5* for astrocyte-3, and *Gfap* and *Map7* for astrocyte-4 (Figure 5B, Appendix A). The DGE analysis between the model and icaritin groups showed that, in astrocyte-0, astrocyte-1, and astrocyte-2, there were significant changes in energy metabolism-related pathways, such as the mTOR or AMPK signaling pathway (Figure 5C and Appendix A) and astrocyte-0 was involved in the phospholipid metabolism, such as the phospholipase D signaling pathway. Using the MALDI–MSI in situ imaging method to detect the levels of metabolites in the substantia nigra of rats in the model or icaritin-treated group, we found that icaritin reversed the changes in the levels of ATP and ADP in the substantia nigra of rats treated with rotenone, indicating that icaritin improved the energy metabolism in PD (Figure 5D). Moreover, the downregulated levels of phosphatidic acid (PA) (18:0/18:1) (X:Y, number of carbon atoms: number of double bonds) was reversed by icaritin treatment, and phosphatidylethanolamine (PE) (16:0/18:1) was also decreased in the substantia nigra of rotenone-administered rats, whereas no changes were detected in the levels of the remaining seven phospholipids (Figure 5D and Appendix A). Phospholipids play important structural and metabolic roles in living cells, participating in the repair and maintenance of the cell membrane [32]. Therefore, these results suggest that icaritin application exerted anti-PD effects by regulating phospholipid homeostasis to promote astrocyte functional recovery. In turn, phospholipase D plays an important role in the synthesis of PA [32]; here, icaritin affected the PA content based on the increased level of glycosylphosphatidylinositol-specific phospholipase D (GPI-PLD) in the substantia nigra of rotenone-administered rats, and had a strong affinity for GPI-PLD (Figure 5E,F, Appendix A), further confirming that the beneficial effect of icaritin in the rat model of PD included phospholipid regulation. In addition, the amino acid metabolism pathways were significantly upregulated in astrocyte-4, which was validated by the observation of a significant change in aspartic acid in the rat substantia nigra between the model and icaritin groups (Figure 5D). Furthermore, we observed that the AMP/ATP ratio was increased in the substantia nigra after icaritin treatment (Appendix A). An increase in the AMP/ATP ratio was previously reported to induce the AMPK signaling pathway for neuroprotection [33,34]; here, the AMPK signaling pathway was highly activated in astrocyte-1 and astrocyte-2 (Figure 5C); thus, the role of astrocytes in the treatment of PD by icaritin was further confirmed. Taken together, these results showed that icaritin significantly changed the pathways of astrocytes in PD, i.e., energy metabolism, phospholipid metabolism, and amino acid metabolism pathways.

### 3.6. Icaritin Treatment Mainly Affects Cytoplasmic Translation and Protein Synthesis Related Pathways in Oligodendrocytes and Oligodendrocyte Progenitor Cells of the Substantia Nigra from Rats with PD

The 6796 OPCs and ODCs were clustered in five separate subsets (Figure 6A). The top markers could be identified using CellRanger for classic marker genes [22], such as *Mal* and *Mcam* for subcluster-0, *Rasgrp3* and *Atp10a* for subcluster-1, *Ptprz1* and *Sox6* for subcluster-2, *Plekha1* and *Fam214a* for subcluster-3, and *Fyn*, *Sema5a*, and *Tnr* for subcluster-4 (Figure 6B, Appendix A). *Pdgfra*^+^ or *Cspg4*^+^ cells were mainly distributed in subcluster-2 (Appendix A), whereas *Mobp*^+^, *Mbp*^+^, or *Mog*^+^ cells were mainly distributed in subcluster-0, subcluster-1, subcluster-3, and subcluster-4 (Appendix A). *Pdgfra* and *Cspg4* were the classic markers of OPCs, whereas *Mobp*, *Mbp*, and *Mog* were the classic markers of ODCs. In addition, through a DGE analysis of OPCs and ODCs in the substantia nigra of the model and icaritin groups of rats, we found that the differentially expressed genes in OPCs were mainly biased toward the positive position of DC1, whereas the differentially expressed genes in ODCs were mainly biased toward the DC1 negative position; in turn, the differentially expressed genes of subcluster-0 and subcluster-1 in OPCs were concentrated in opposite directions of DC2 (Figure 6C). Furthermore, there were significant differences in cytoplasmic translation and protein synthesis pathways in subcluster-0, subcluster-1, subcluster-2, and subcluster-3 between the model and icaritin-treated groups. Moreover, the synapse organization pathway in subcluster-0 was prominently changed. Of note, the pathways related to the cell cycle in subcluster-4 were significantly different, suggesting a change in cell fate after the treatment (Figure 6D and Appendix A). These results demonstrated that OPCs/ODCs were involved in the development of PD in rats and were affected by icaritin treatment.

### 3.7. Icaritin Treatment Modulates Energy Metabolism and Oxidative Stress of Microglia and Endothelial Cells in the Substantia Nigra of Rats with PD

Brain microglia can act as regulators of neuronal function and homeostasis in the adult brain; moreover, chronic impairment of the microglia–neuron cross-talk may lead to the permanent failure of synaptic and neuronal function and the health of patients with PD [35]. In this study, microglia were clustered in four subsets (Appendix A). The top markers could be identified using CellRanger for classic marker genes [22], such as *Nav3* and *Lrp4* for microglia-0, *Mrc1* and *Cd163* for microglia-1, *Skap1* and *Ccl5* for microglia-2, and *Pak7* and *Tmeff2* for microglia-3 (Appendix A). Based on Gene Ontology analysis and comparison between the model and icaritin groups, the expression of several genes in PD-related pathways, including the succinate dehydrogenase complex iron sulfur subunit B (*Sdhb*) gene, NADH: ubiquinone oxidoreductase subunit A (*Ndufa7* and *Ndufa8*), and calcium/calmodulin-dependent protein kinase II delta (*Camk2d*), were significantly altered in microglia (Appendix A), indicating energy metabolism and oxidative stress-related pathways were strongly activated.

In addition, ECs were clustered in three subsets (Appendix A), wherein the top markers, such as *Bmp6* and *Ptgds* for EC-0, *Cldn5* and *Cxcl12* for EC-1, and *Ebf1* and *Pde8b* for EC-2, were identified (Appendix A). Similarly, Gene Ontology analysis after icaritin treatment revealed that the expressions of most genes in the ATP metabolic process were promoted, including NADH: ubiquinone oxidoreductase subunit (*Ndufs7*, *Ndufb6*, *Ndufb9*), forkhead box O3 (*Foxo3*), and lactate dehydrogenase A gene (*LDHA*), suggesting the upregulation of oxidative stress and promotion of endothelial cell metabolism in ECs. The expression of most genes in the Wnt signaling pathway was downregulated, including the cell cycle progression-related gene *CDK14* and cell proliferation and development-related gene *Wnt4* (Appendix A). These results suggest that microglia and ECs are involved in the onset of PD and that the energy metabolism and oxidative stress-related genes are significantly altered after icaritin treatment.

## 4. Discussion

In the present study, we assessed the efficacy of the flavonoid icaritin in the treatment of PD. We revealed that icaritin significantly improved the exercise capacity in a rat model of PD by inhibiting DA and serotonin degradation. Moreover, scRNA-seq was first applied to identify a new synapse-like cell cluster, SRCs, in the substantia nigra of rats with rotenone-induced PD. Notably, the synapse-related signaling functions of this cluster improved after icaritin treatment. Furthermore, icaritin treatment promoted the recovery of the physiological homeostasis of neurons, astrocytes, OPCs, ODCs, microglia, and ECs. Collectively, our study results provided evidence that icaritin plays a therapeutic role in rats with rotenone-induced PD by elucidating its regulatory effects on different cell populations in the substantia nigra.

Parkinson’s disease (PD) is the second most common neurodegenerative disease with complex pathogenesis. The pathogenesis of the rotenone-induced model of PD is similar to that of human PD, which replicated the inclusion bodies in PD dopaminergic neurons and oxidative stress damage concomitantly; thus, the rotenone model summarized the most important mechanisms in PD pathogenesis [2,14]. The natural flavonoid glucoside extracted from *E. sagittatum* Maxim., icaritin, which is a metabolite of icariin in the human intestinal microflora, confers neuroprotection in mice with PD by simultaneously reducing neuroinflammation and oxidative stress and improving alterations in neuronal energy metabolism [11,36]. Interestingly, the prescription medication of *E. sagittatum* Maxim. or icariin has been widely used in the clinical treatment of benign prostatic hyperplasia and sexual dysfunction, which is a common but under-recognized nonmotor feature in PD [37,38]; however, research on the use of benign prostatic hyperplasia therapeutic drugs for the management of PD has only recently been reported. Terazosin and related quinazoline agents slow disease progression, reduce PD-related complications in individuals with PD, and decrease the risk of PD diagnosis [39,40]. Terazosin acts by increasing the activity of phosphoglycerate kinase 1, which is a key ATP-generating enzyme in the glycolytic pathway. Although terazosin and icaritin have different molecular structures, they can treat PD by improving energy metabolism, thus possibly opening new research directions for the discovery of anti-PD drugs.

Subsequently, we applied scRNA-seq to derive a detailed transcription pattern of cells in the substantia nigra of rats with PD after icaritin treatment and identify a new synapse-like cell cluster, SRCs, in which genes related to synaptic function, signal transmission, synaptic structure, synaptic vascular endotheliosis, and other pathways were highly upregulated. Additionally, icaritin increased cell numbers and was strongly bound to transcription factors related to synaptic function in SRCs, e.g., SMAD3, CEBPB, and NR3C2. These transcription factors or cofactors might be regulated by icariin, which can be metabolized into icaritin. SMAD3 is associated with neurodevelopment, dendrite growth, and synaptogenesis [41,42], and icariin inhibits airway remodeling by attenuating the TGF-β1-induced epithelial–mesenchymal transition by targeting Smad and MAPK signaling [43]. CEBPB regulates the expression of genes involved in immune and inflammatory responses to escalate Alzheimer’s disease-related gene expression and pathogenesis [44]. Conversely, icariin significantly increases ALP activity and downregulates PPARG, adipsin, and CEBPB genes during adipogenic differentiation [45,46]. Furthermore, NR3C2 is one of the Alzheimer’s disease genes targeted by active compounds of the Bushen Tiansui Formula, such as icariin [47,48]. Therefore, icariin or its metabolite, icaritin, may regulate these transcription factors or cofactors to attenuate neuroinflammation and oxidative stress in PD rats.

Furthermore, we revealed that SRCs had apparent interactions with neuron clusters via the synaptic signal transmission-related genes *Nlgn1* and *Nlgn3* and the synaptic junction-related genes *Nrxn1*, *Nrxn2*, and *Nrxn3*. The adhesion system formed by neurexins and neuroligins bidirectionally orchestrates the function of presynaptic and postsynaptic terminals [49]. Nguyen et al. highlighted the disruptions in synaptic vesicle endocytosis as a significant contributor to PD pathogenesis [50]. Synaptic damage in PD preceded neuronal degeneration and revealed that synaptic energetic failure and accumulation of dysfunctional organelles occur sequentially as the earliest PD events at the dopaminergic terminals to accelerate neuronal degeneration [51]. Therefore, we suspected that the abnormal function and population of SRCs in the substantia nigra cause neural network disorder, impede signal transmission between cells, and aggravate PD symptoms. Notably, the protective effect of icaritin on SRCs may represent a new mechanism for treating PD.

During PD progression, the key roles of neurons, astrocytes, OPCs, ODCs, microglia, and ECs have been clarified [52,53,54,55,56,57]. Eight subgroups of neurons were identified, with the neuron-5 subcluster comprising dopaminergic neurons, which is inconsistent with a previous study [23]; further, the gene encoding tyrosine hydroxylase was not highly expressed, whereas *Ak5*, *Ptprd*, *Grm3*, and *Pdzd2* showed high expression. The PI3K/AKT signaling pathway is activated in dopaminergic neurons and involved in regulating the production and release of neurotrophic factors to induce inflammation [25,26]. Icaritin may enhance the survival and maturation of dopaminergic neurons and promote significant expression of genes related to neural development and exocytosis in neurons, which downregulate genes related to neuroinflammation and are conducive to PD recovery. These findings provide the basis for the anti-neuroinflammatory effects of icaritin on a cellular subset of the substantia nigra. In the adult brain, astrocytes and microglia are essential for maintaining the neuronal environment and are involved in several processes, such as the recirculation of neurotransmitters from the synaptic cleft, maintenance of the blood–brain barrier, and regulation of energy homeostasis [58]. Icaritin treatment also significantly affects phospholipid metabolism, energy metabolism-related and amino acid metabolism pathways in astrocytes, as well as energy metabolism and oxidative stress in microglia, which have been reported to be disturbed during PD [11]. Moreover, the MALDI-MSI results partially verified the molecular pathway changes in astrocytes. Additionally, in OPCs and ODCs, the pathways related to the cell cycle, such as the cytoplasmic translation and protein synthesis pathways, were significantly altered after icaritin treatment. Finally, icaritin application significantly regulated energy metabolism and oxidative stress-related genes of microglia and ECs involved in the onset of PD. Specifically, Wnt signaling pathways related to PD formation in ECs [59] were altered by icaritin. Thus, the single-cell analysis showed that icaritin supplementation attenuated neuroinflammation, oxidative stress, and energy deficiency in PD, and the therapeutic effect of icairitin is extensive, effective, and exploitable.

## 5. Conclusions

In conclusion, our current data reveal the key role of icaritin in improving the exercise capacity in a rat model of PD by inhibiting DA and serotonin degradation. More importantly, we first apply scRNA-seq to identify a new synapse-like cell cluster, SRCs, in the substantia nigra of PD rats, in which the synapse-related signaling functions were improved after icaritin treatment. Icaritin application also promoted the recovery of the physiological homeostasis of neurons, astrocytes, OPCs, ODCs, microglia, and ECs. Consequently, icaritin supplementation may emerge as a potential novel diagnostic and therapeutic strategy in PD as well as other neurodegenerative diseases.

## 6. Limitations of the Study

There are several limitations of the study that should be acknowledged. First, the mechanism by which SRCs play a role in PD was not fully elucidated, and the regulation by icaritin in SRCs warrants further in-depth studies. In addition, the MALDI-MSI analysis results did not fully indicate the pathway changes in astrocytes in single-cell sequencing. Moreover, genetic cell-lineage tracing studies in rat may more directly demonstrate the anti-PD effect of icaritin application on different cell clusters. Overall, our study provides an important basis for the treatment of PD with icaritin.

## Figures and Tables

**Figure 1 antioxidants-13-01183-f001:**
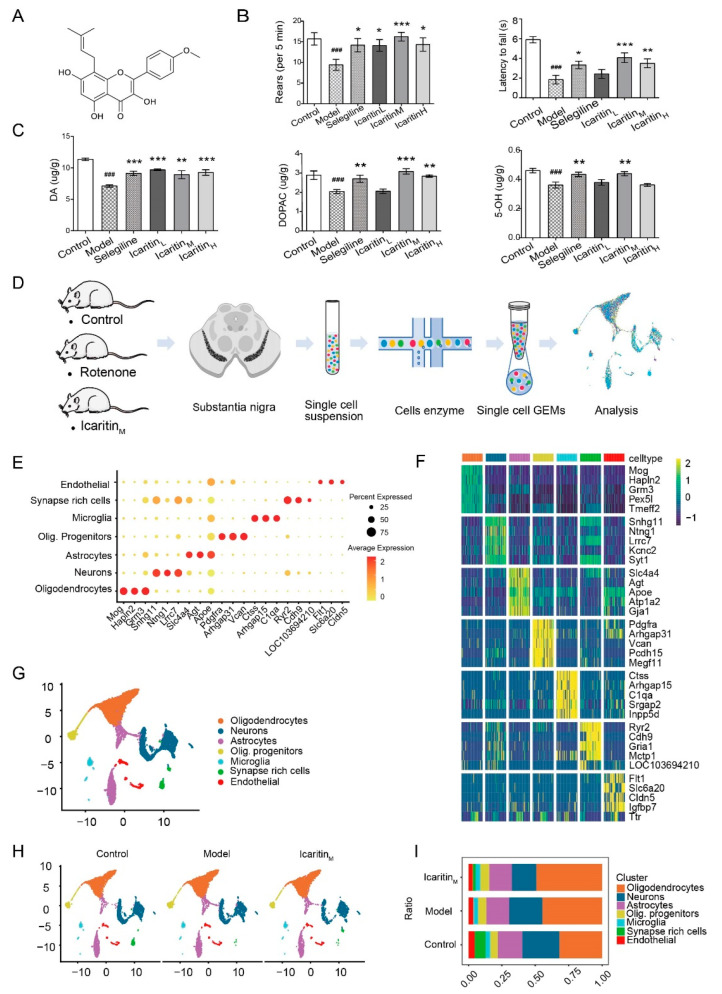
Single-cell transcriptome atlas of the substantia nigra cells in rotenone-induced PD rats. (**A**). Chemical structure of icaritin. The molecular weight is 368.38 atomic mass units, and the chemical name is 3-[(5,7-dihydroxy-4-phenyl-chromenyl)oxy]-3-methyl-1-buten-4-one. (**B**). The rearing behavior test (5 min) and wire grip test in rats. Data are presented as the mean ± SEM. *n* = 9–10. (**C**). HPLC analysis of DA, DOPAC, and serotonin levels. Data are presented as the mean ± SEM. *n* = 4–5. (**D**). Schedule of single-cell transcriptome atlas in the substantia nigras from control, model, and icaritin groups. (**E**). Dot plots showing the 21 signature gene expressions across the 7 cellular clusters. The size of dots represents the proportion of cells expressing the marker, and the spectrum of color indicates the mean expression levels of the markers (log1p transformed). (**F**). Row-normalized single-cell gene expression heatmap of cell-type marker genes. (**G**). UMAP plot of all cells clustered and color coded by cell type. (**H**). The UMAP plot shows the changes in cell types under different treatments. (**I**). Relative proportion of each cell cluster across 3 groups as indicated. The values of the detailed relative proportion of each cell cluster are provided in the Source Data file. Control, control group; Model, PD model group; Selegiline, selegilin-treated group; Icaritin_L_: 3.27 mg/kg icaritin-treated group; Icaritin_M_: 6.54 mg/kg icaritin-treated group; Icaritin_H_: 13.08 mg/kg icaritin-treated group, ^###^ *p* < 0.001 vs. Control group; * *p* < 0.05, ** *p* < 0.01, *** *p* < 0.001 vs. Model group.

**Figure 2 antioxidants-13-01183-f002:**
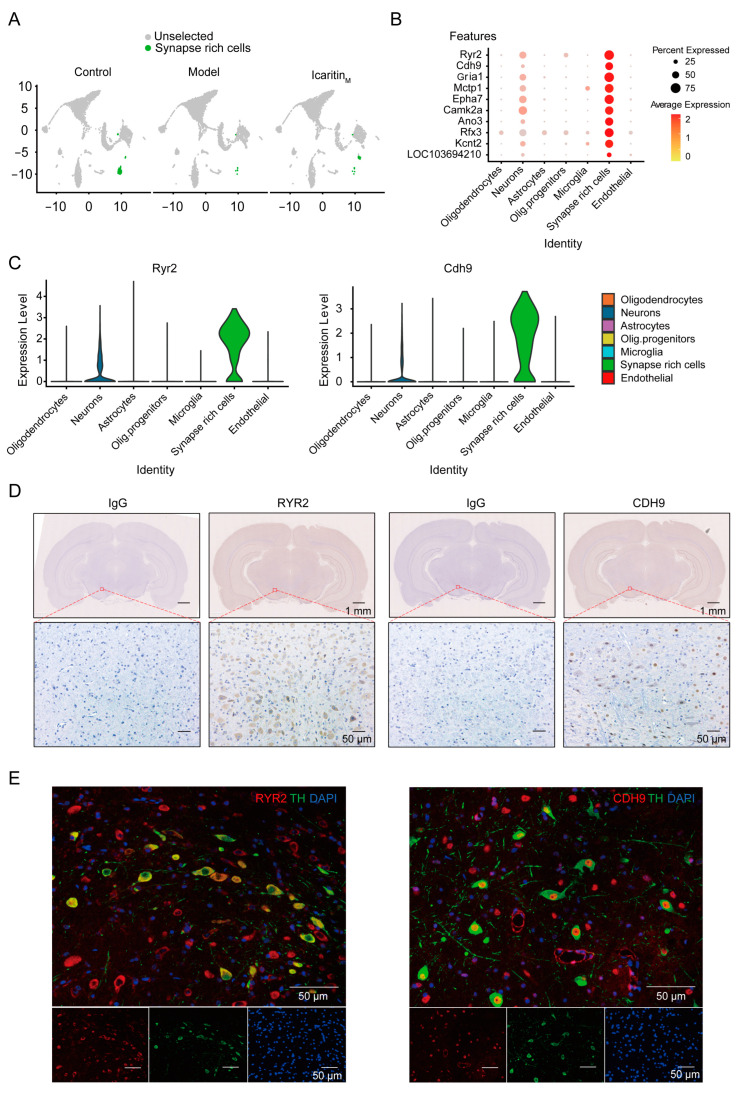
Cells that characterize specific synapse-related genes are significantly downregulated in the substantia nigra tissue of PD rats. (**A**). The UMAP suggested the proportion of SRCs among three groups. (**B**). Dot plots showing the top 10 marker gene expressions of SRCs across the 7 cellular clusters. (**C**). A violin plot showing the Ryr2 and Cdh9 marker genes of SRCs across the 7 cellular clusters. (**D**). Representative images of RYR2 (**left**) and CDH9 (**right**) expression by immunohistochemistry staining in rat substantia nigra. The scale bar has been annotated in the diagram. Immunohistochemistry was performed with three animals per group. (**E**). Representative images of RYR2 (**left**) and CDH9 (**right**) expression by immunofluorescence staining in rat substantia nigra and costaining with tyrosine hydroxylase and DAPI. The scale bar has been annotated in the diagram. The experiments were repeated three times. (**F**). Dot plot heatmap showing GO biological process terms enriched in SRC marker genes. Hypergeometric test for overrepresentation; Benjamini–Hochberg multiple test correction. (**G**). Emapplot showing clusters of Gene Ontology biological process terms enriched in SRC marker genes. Hypergeometric overrepresentation test, Benjamini-Hochberg multiple testing correction. (**H**). The tree plot showing clusters of KEGG process terms enriched in SRC marker genes. Control, control group; Model, PD model group; Icaritin_M_: 6.54 mg/kg icaritin-treated group.

**Figure 3 antioxidants-13-01183-f003:**
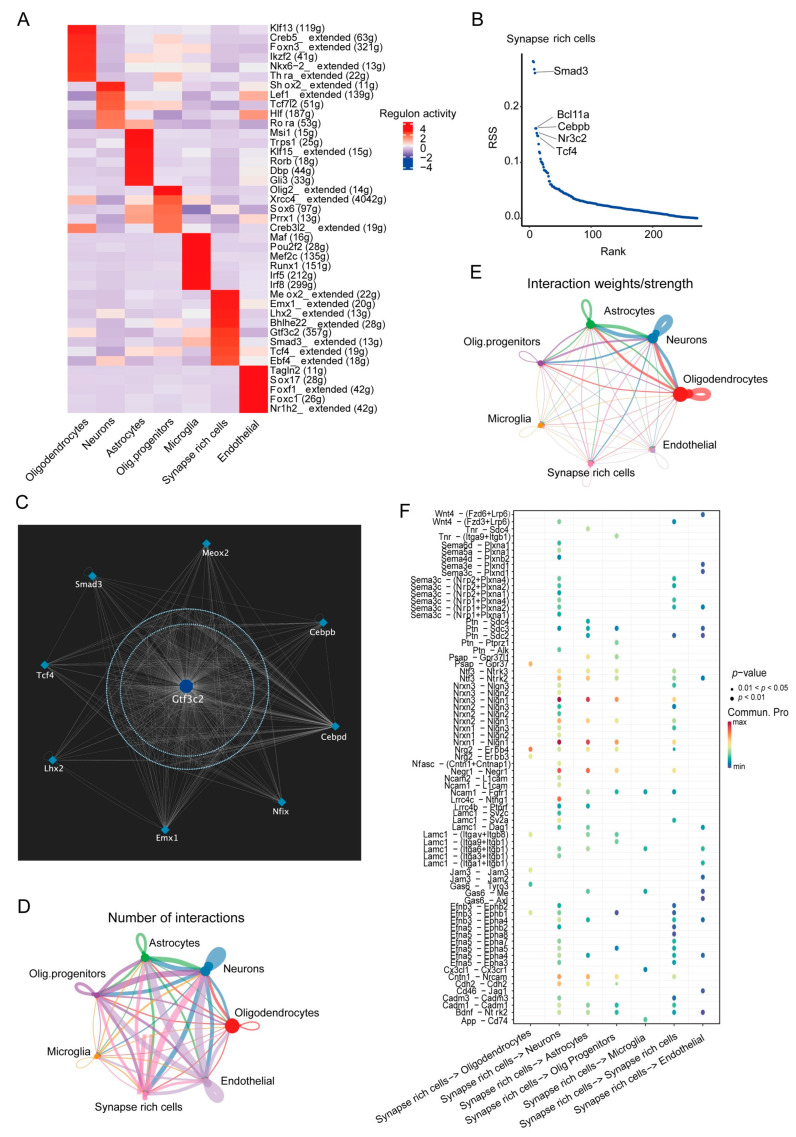
SRCs interact tightly with other cell populations in the substantia nigra of PD rats. (**A**). Heatmap of the *t* values of AUC scores of expression regulation by transcription factors (TFs), as estimated using SCENIC, per cell type. (**B**). The top five regulons (TFs and corresponding target gene candidates) in SRCs. (**C**). A circle plot showed TF score relation network in SRCs. Shown are nodes (TFs) and target genes. (**D**,**E**). A circle plot showing the interactions among cell types across all samples regarding the number (**left**) and the weight/strength of the interactions (**right**). (**F**). Bubble plots of the main signaling pathways from SRCs to all cell groups derived from Cellchat. (**G**). Circos plots showing main signaling pathways from all cell groups to SRCs derived from Cellchat. (**H**). Circos plots of the main signaling pathways from SRCs to all cell groups derived from Cellchat.

**Figure 4 antioxidants-13-01183-f004:**
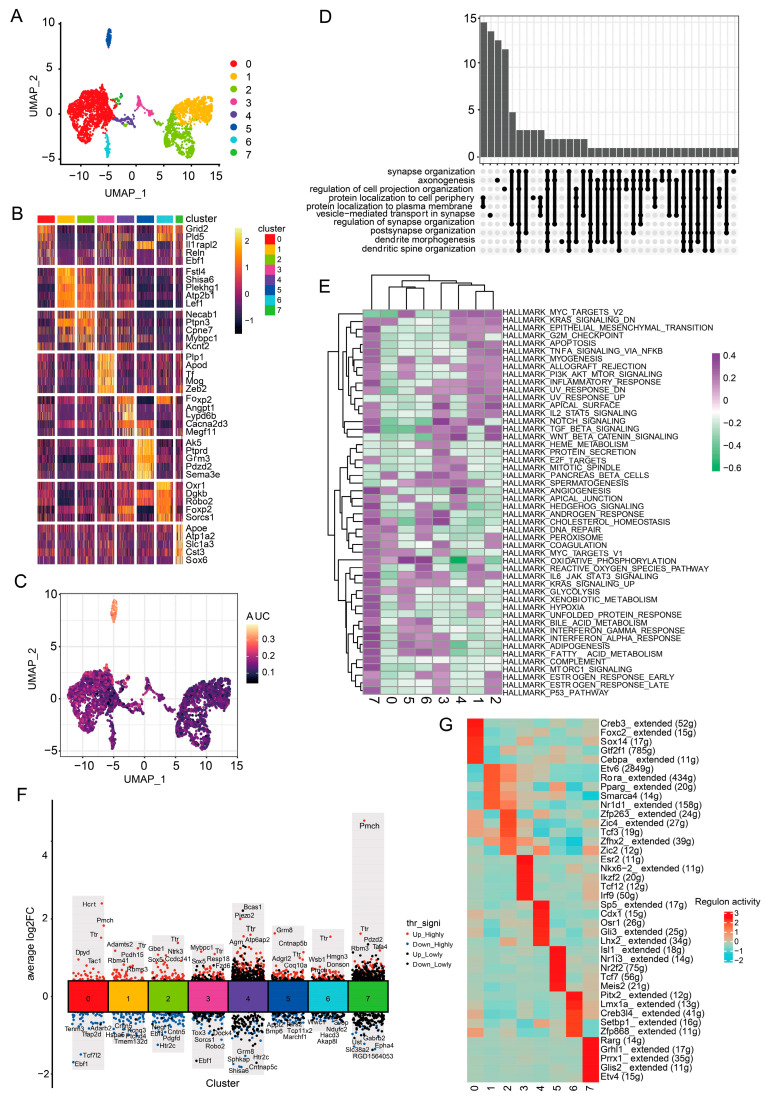
Icaritin treatment improves protein localization and synaptic vesicle-mediated transport-related pathways in neurons of the substantia nigra from rats with PD. (**A**). UMAP plot of 3282 neurons, color-coded by their associated cluster. (**B**). Row-normalized single-cell gene expression heatmap of cell-type marker genes. (**C**). Individual cell AUC score overlay for dopamine neurons geneset activities. (**D**). The heatmap of GSVA of the 50 hallmark gene sets in MSigDB database among the eight neuron cell subclusters. (**E**). The tree plot showing clusters of KEGG process terms enriched in per cell subtype. (**F**). The differentially expressed genes (DEGs) among the cell population of the Icaritin samples compared to Model. The labeled genes were the top ten upregulated or downregulated DEGs ranked by the average log2FC. (**G**). Heatmap of the t values of AUC scores of expression regulation by transcription factors (TFs), as estimated using SCENIC, per cell subtype.

**Figure 5 antioxidants-13-01183-f005:**
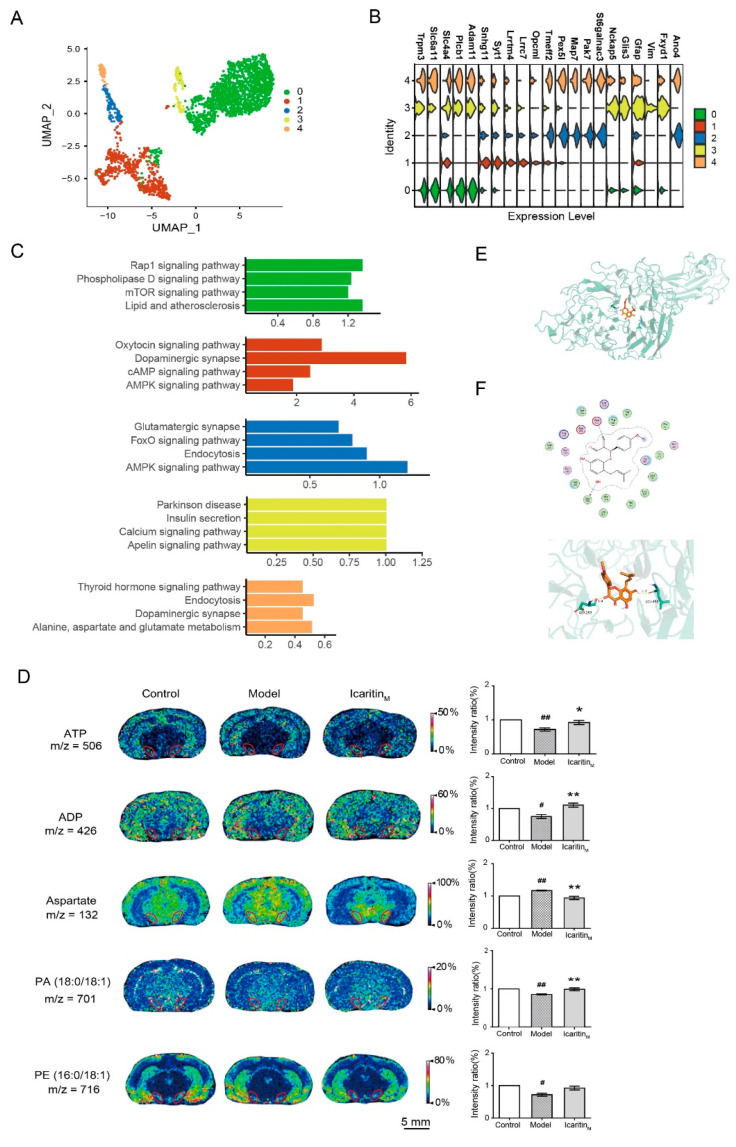
Amelioration of icaritin on the energy metabolism, phospholipid metabolism, and amino acid metabolism-related pathways in astrocytes of the substantia nigra from rats with PD. (**A**). Five main 2462 astrocyte cell subclusters were identified by UMAP analysis. (**B**). Violin plots showing the expression distribution of selected canonical cell markers in the 5 clusters. The rows represent selected marker genes, and the columns represent clusters of the same color as in (**A**). (**C**). KEGG analysis showing enriched terms in each indicated gene cluster using the differentially expressed genes (DEGs) among the cell population of the icaritin samples compared to Model. (**D**). In situ MALDI–MSI of ATP, ADP, aspartate, PA (18:0/18:1), and PE (16:0/18:1). Spatial resolution: 200 μm; scale bar: 5 mm; *m*/*z*: mass-to-charge ratio. The area selected by the red line is substantia nigra. *n* = 3. (**E**). Docking result of icaritin with GPI-PLD. The orange molecule is icaritin. (**F**). Location of icaritin’s binding site on GPI-PLD (site 1 in Appendix A). Control, control group; Model, PD model group; Icaritin_M_: 6.54 mg/kg icaritin-treated group. ^#^ *p* < 0.05, ^##^ *p* < 0.01 vs. Control group; * *p* < 0.05, ** *p* < 0.01 vs. Model group.

**Figure 6 antioxidants-13-01183-f006:**
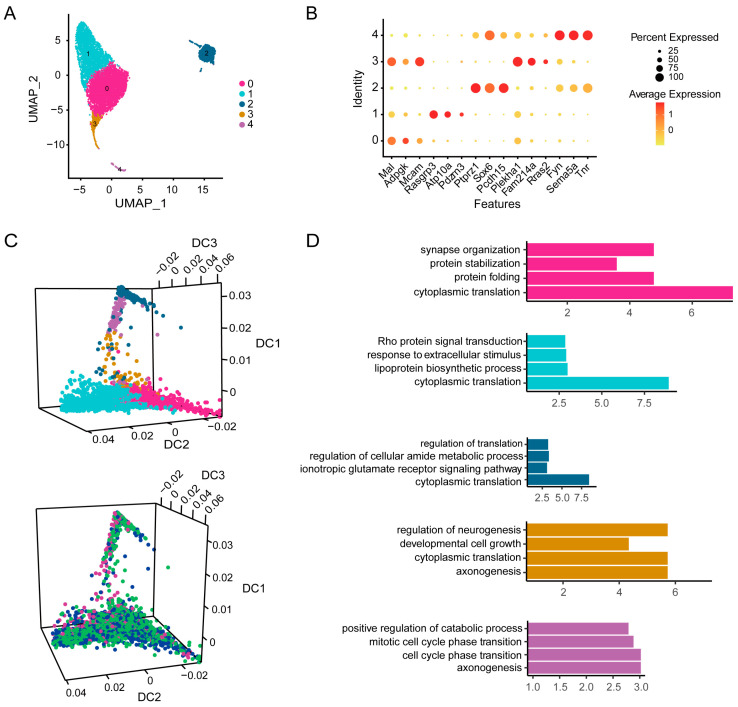
Improvement of icaritin on the gene expressions of oligodendrocytes and oligodendrocyte progenitor cells in the substantia nigra of rats with PD. (**A**). UMAP plot of 5936 oligodendrocytes and 860 oligodendrocyte progenitor cells, color-coded by their associated cluster. (**B**). Dot plots showing the 15 signature gene expressions across the 5 cellular clusters. (**C**). Diffusion component (D.C.) analysis of oligodendrocytes and oligodendrocyte progenitor cells colored by subcluster. (**D**). GO analysis showing enriched terms in each indicated gene clusters using the differentially expressed genes (DEGs) among the cell population of the icaritin group compared to the model group.

**Table 1 antioxidants-13-01183-t001:** Results of molecular docking analysis.

ID	Molecules	Full Name of Target	Source	Uniprot ID	Docking Score (Kcal/mol)
1	Icaritin	B cell CLL/lymphoma 11A (zinc finger protein)	Rat	D3ZSY3	−8.56
2	Icaritin	CCAAT/enhancer-binding protein beta	Rat	P21272	−6.59
3	Icaritin	Mineralocorticoid receptor	Rat	P22199	−10.21
4	Icaritin	Mothers against decapentaplegic homolog 3	Rat	P84025	−9.51
5	Icaritin	Transcription factor 4	Rat	Q62655	−6.15

## Data Availability

The data presented in this study are available within the article and in Appendix A.

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
