# Peer review of "A Single-Cell Atlas of the Substantia Nigra Reveals Therapeutic Effects of Icaritin in a Rat Model of Parkinson’s Disease"

_antioxidants, 2024, doi:10.3390/antiox13101183_

Round 1

Reviewer 1 Report

The paper entitled "A single-cell atlas of the substantia nigra reveals therapeutic effects of icaritin in a rat model of Parkinson's disease" is a very interesting study. The use of novel antioxidant molecules in conditions such as Parkinson's disease (PD) or other neurodegenerative diseases is critical to understanding the underlying molecular mechanisms and improving patient outcomes.

I would like to commend the authors for their work and offer the following minor suggestions:

Literature review: It would be beneficial to expand the literature review on icaritin in the introductory section. This would provide the reader with a better understanding of the known effects of icaritin as reported in the existing literature.

Figure 2 Scale Bar: Please consider including the scale bar from Figure 2 in the figure caption.

Reviewer 2 Report

In the current manuscript, authors have utilized scRNAseq to track the changes in cell population of substantia Niagara following treatment with Icaritin a flavonoid metabolite in a rotenone induced PD model in SD rats. The manuscript is well written, and the techniques explained clearly. However, I have few questions and suggestions.

1.  What do the authors mean by administered intragastrically? How were the treatments administered. Were the treatment administered daily.

2.       It would be helpful if the authors provided the timeline of experiment in Figure 1. I understand it has been provided in the supplementary figures but in my opinion, it can be moved to Figure 1 for better visibility. Also, the timeline depicted in schematic and the description in the methods section do not match. For instance, in the text, the authors mention that the behavioral studies were carried out one day after the last injection. In the schematic however it is at the end of 25 days. Please clarify.

3.       Please state exactly how many animals were used for each sub experiments. For instance, for the RNAseq analysis, were the substantia nigra analyzed separately from each animal. Were the studies repeated and the data finally presented as mean changes.

4.       Figure 1: Please indicate the molecular wight of the compound and if available the chemical name of the compound.

5.    Why do the authors think that the higher doses of Icaritin not beneficial in this model. According to the authors in their previous publication; https://doi.org/10.3390/antiox10040529 demonstrated that a higher Icaritin dose was beneficial.

6.     According to the Figure 1: (I) the neuronal gene expression is the least in animals treated with Icaritin and the authors still seem to see a beneficial effect. Please clarify.

7.    The authors claim that the beneficial effects can be attributed to a previously unknown cluster namely Synapse Rich Cells (SRC). They further suggest that these are non-dopaminergic cells as they are not positive for tyrosine hydroxylase. What do the authors propose these cells are? Neurons, astrocytes membrane/synaptic vesicles, heterogenous cell cluster with similar enrichment of the genes or dying cell population. Is it possible to further characterize these cell types by selectively isolating them from tissues. How does this cluster differ from previously reported RNA seq studies and what was the difference in analysis performed by the authors that led them to identify this cluster.

8.   Figure 1 (I). Does Selegiline increase the SRCs. This data can further confirm that the protection ion in PD can be attributed to SRC’s.

9.     Figure 1 (B). Dopamine and serotonin levels in response to Selegiline demonstrated in figures but not included in the methods section.

10.   Panels in Figure (E) can be improved. DAPI is not visible and therefore synapse rich cells are not clear.

11.   The texts describing the cell-interaction and molecular docking studies are too long and does not convey the results clearly.

12.   If SRC is not a distinct cell type does cell-cell interaction studies add any value? Please clarify

13.   The authors suggest Icaritin to modulate transcription factors including SMAD3, CEBP and cite articles where Icaritin has been shown to regulate these transcription factors. Can the authors confirm if it is a positive or a negative regulation of these transcription factors?

14.   What is the proposed mechanism of action for Icaritin. The authors propose a decrease in degradation of dopamine and serotonin. While they have demonstrated the increase there is no data to show that Icaritin prevent the degradation of these neurotransmitters.

15.   The study design is prophylactic. Can we see the protective effects of Icaritin after detecting PD symptoms?

16.   Figure 5D. How do the ATP, ADP aspartate, PA and PE compare to Icaritin vs Selegiline treatment?

NONE

Reviewer 3 Report

The study from Wu et al provides interesting results into the cellular complexity within the substantia nigra of a rotenone-induced rat model of Parkinson’s disease (PD). The work highlights the interaction between various cell populations, including a newly identified subtype: synapse-rich cells (SRCs). The work focuses in understanding the selective vulnerability of dopaminergic neurons in PD by using single-cell RNA sequencing to delineate seven distinct cell clusters, thus contributing to the growing body of research on cellular heterogeneity in neurodegeneration. The finding that icaritin, a flavonoid compound from a well-known plant used in Chinese medicine, protects cells against rotenone is promising.

Major comments:

1. In the table 1 you show icaritin docking results but the validation of some of these  targets should be addressed. In the sameline, the study shows that icaritin binds to transcription factors in SRCs, the exact molecular mechanisms by which icaritin exerts its protective effects remain unclear. It is needed to propose and to demonstrate (it could be with neuronal cell cultures of human dopaminergic cells such as SH-SY5Y or other cell lines including astrocytes) what is the specific mechanism of action of icaritin.

2. While the study identifies SRCs as a new cell subtype, further functional validation is essential to understand their specific role in PD.3. 

3. MAO B increased activity is suggested to be playing a key role in PD onset and progression. Is there any regulatory effect of icaritin on MAO B expression and /or activity

Minor comments

1.  Cell survival assays with increasing concentrations of icaritin should be performed to identify the optimal concentrations range.

2. You made comments on the role of oxidative stress and neuroinflammation in PD. May be you would analyze the antioxidant effect of icaritin in vitro with cell lines exposed to oxidative stress. Regarding neuroinflammation that is mainly carried out by microglia some comments on the role of these cells in the result section or discussion should be included.

3. Please explain to the wide variety of readers of this journal what the latency in Figure 1B means and why the treatments with icaritin do not produce any regulation of it compared to the rotenone treatment.

4. Explain how the reduction of astrocyte metabolism contribute to PD onset and progression (and how its recovery with icaritin is benefitial).

Round 2

Reviewer 3 Report

No comments for authors.

No comments for authors.
